# Rapidly Growing Protein-Centric Technologies to Extensively Identify Protein–RNA Interactions: Application to the Analysis of Co-Transcriptional RNA Processing

**DOI:** 10.3390/ijms22105312

**Published:** 2021-05-18

**Authors:** Akio Masuda, Toshihiko Kawachi, Kinji Ohno

**Affiliations:** Center for Neurological Diseases and Cancer, Division of Neurogenetics, Nagoya University Graduate School of Medicine, Nagoya 466-8550, Japan; t0asdvb@hotmail.com (T.K.); ohnok@med.nagoya-u.ac.jp (K.O.)

**Keywords:** CLIP, tRIP, RNA polymerase II, RNA-binding protein, co-transcriptional RNA processing

## Abstract

During mRNA transcription, diverse RNA-binding proteins (RBPs) are recruited to RNA polymerase II (RNAP II) transcription machinery. These RBPs bind to distinct sites of nascent RNA to co-transcriptionally operate mRNA processing. Recent studies have revealed a close relationship between transcription and co-transcriptional RNA processing, where one affects the other’s activity, indicating an essential role of protein–RNA interactions for the fine-tuning of mRNA production. Owing to their limited amount in cells, the detection of protein–RNA interactions specifically assembled on the transcribing RNAP II machinery still remains challenging. Currently, cross-linking and immunoprecipitation (CLIP) has become a standard method to detect in vivo protein–RNA interactions, although it requires a large amount of input materials. Several improved methods, such as infrared-CLIP (irCLIP), enhanced CLIP (eCLIP), and target RNA immunoprecipitation (tRIP), have shown remarkable enhancements in the detection efficiency. Furthermore, the utilization of an RNA editing mechanism or proximity labeling strategy has achieved the detection of faint protein–RNA interactions in cells without depending on crosslinking. This review aims to explore various methods being developed to detect endogenous protein–RNA interaction sites and discusses how they may be applied to the analysis of co-transcriptional RNA processing.

## 1. Introduction

Following transcription by RNA polymerase II (RNAP II), nascent RNA undergoes processing events of a 7-methyl guanosine cap at the 5′ end (5′ capping), splicing of exons, and formation of the 3′ end by cleavage and polyadenylation, to produce mature mRNA (Figure 1). The specific mRNA processing machinery, such as the capping enzyme complex, spliceosome, and 3′ end processing machinery are recruited to transcribe RNAP II and initiate co-transcriptional mRNA processing [1]. Recent studies have revealed a close relationship between mRNA transcription and processing [1,2]. For example, slow and rapid transcription elongations facilitate and suppress splicing, respectively [2]. In contrast, the binding of splicing factors, such as U1 snRNP [3], SRSF2 [4], FUS [5], and SFPQ [6], to pre-mRNA affects transcription elongation in addition to the processing of nascent RNA. The dynamic phosphorylation pattern of the C-terminal domain (CTD) of RNAP II plays an important role in the regulation of the transcriptional status of RNAP II [7], and active spliceosomes are complexed, especially to serine 5 phosphorylated CTD, during elongation [8]. This represents the physiological connection between RNAP II and mRNA processing. In the RNA processing machinery, direct interactions are formed between the RNA-binding proteins (RBPs) and RNA, which disappear soon after the completion of RNA processing. The transient nature of protein–RNA interactions complicates their experimental identification.

A recent study detected more than 1000 proteins, which are directly associated with RNA in human cells [9]. Hundreds of these proteins are known to participate in the regulation of mRNA processing and form complex networks composed of proteins and RNAs [8,10,11]. A given RBP is often distributed in a wide range of RNA machineries and is not restricted to the transcription machinery [12]. Therefore, the labeling or isolation of protein–RNA interactions specific to the transcribing RNAP II machinery is necessary to analyze co-transcriptional RNA processing, which further increases the difficulty of the experiments.

As the characterization of physiological protein–RNA interactions is essential to understand how mRNA is transcribed and processed in cells, various methods have been developed to detect such transient and faint interactions, which has been reviewed in several articles [13,14,15]. These methods can be classified into protein-centric and RNA-centric methods. The protein-centric methods generally rely on the purification of a protein of interest, followed by sequencing of the associated RNA. In contrast, the RNA-centric methods capture RNA of interest and identify the associated proteins. This review explores the development of protein-centric methods to identify protein–RNA interaction sites in endogenous RNA and discusses their potential application to the analysis of co-transcriptional RNA processing.

## 2. General Methods to Identify Protein–RNA Interaction Sites in RNA

### 2.1. Development of Methods to Detect Protein–RNA Interaction Sites

It had long been difficult to obtain a transcriptome-wide map of direct protein–RNA interaction sites before the development of high-throughput sequencing technologies. Instead, in vitro identified binding motifs of RBPs were searched in a specific RNA sequence to presume protein–RNA interaction sites. The consensus binding motifs of RBPs were derived using several methods such as SELEX [16], RNAcompete [17], and RNA Bind-N-Seq [18], in which sequences of in vitro bound RNAs by the distinct recombinant RBPs were extracted. A score matrix of each RBP binding motif was created by collecting these sequences and utilized in several services, including ESE finder [19] and SpliceAid [20], to predict the potential binding sites of RBPs in a given sequence. These analyses have helped to shed light on the complex networks of protein–RNA interactions. However, there are certain limitations that prevent the broad application of the obtained results. Most of the consensus binding motifs of RBPs are short and degenerative, resulting in an increase in false positive rates. Even the functionally distinct RBPs often share the same binding motifs. Although these RBPs may bind to the RNA target sites in a spatiotemporally differentiated manner, the analysis does not show when and where the RBPs bind in the cells. Therefore, the predicted interactions need to be further validated using in vivo experiments.

The RNA immunoprecipitation (RIP) method and its variations have been developed to detect physiological protein–RNA interactions. In the primary RIP method, an RBP is immunoprecipitated from the cell lysates using a specific antibody, preserving the protein–RNA complexes formed in cells as much as possible [21,22]. Then, the co-immunoprecipitated RNAs are isolated and assessed with RT-PCR [22], microarray [23,24], and later by high-throughput sequencing [25,26]. Owing to the simplicity of its protocol, the RIP method has been applied to a wide variety of studies, although the obtained results have several limitations. First, it detects the RNA bound directly and indirectly by RBPs, as multiple protein complexes are retained in the procedure. Second, protein–RNA interactions are reconstituted in vitro, to some extent, during the incubation of the cell lysate with the antibody. The obtained results often represent the in vitro binding affinity of an RBP to the target RNA rather than the direct in vivo interaction. These reconstitutions cause a high background, which may obscure the faint in vivo protein–RNA interactions formed co-transcriptionally. Third, the RIP method isolates an entire RNA molecule interacting with an RBP and does not specify the RBP-binding sites in the RNA. On beads, the partial digestion of RNA during immunoprecipitation (IP) effectively works to narrow down the segments bound by RBPs [27], although the other problems described above still remain.

To specifically detect the in vivo direct protein–RNA interaction sites, the chemical crosslinking of the interactions was combined with the RIP method. The crosslinking of protein–RNA interactions enables the elimination of non-specifically remaining RNA and proteins from the immunoprecipitants by stringent washing of protein–RNA complexes with high-salt buffer and ionic detergents, as well as by gel purification. UV irradiation [28], formaldehyde [29], psoralen [30], and methylene blue [31] are known crosslinkers of protein–RNA interactions. Among them, UV irradiation and formaldehyde-crosslinking (FA-crosslinking) have been used to analyze protein–RNA interactions in living cells [29,32]. UV irradiation is advantageous as it does not crosslink a protein–protein interaction but induces the crosslinking of a protein–nucleotide interaction, by generating a covalent bond between these molecules [33]. Thus, UV-crosslinking enables identification of direct protein–RNA/DNA interaction sites, although its crosslinking efficiency of protein–RNA interactions in living cells is only ≈5% [34]. In contrast to UV-crosslinking, FA-crosslinking efficiently induces protein–protein crosslinking, in addition to protein–RNA/DNA crosslinking, resulting in the purification of a large complex including multiple proteins and nucleic acids in the IP step. Therefore, FA-crosslinking might be more suitable for the analysis of RNA machinery than for a single RBP, as seen in the analysis of U1 snRNP [11].

### 2.2. Crosslinking and Immunoprecipitation (CLIP)

Currently, UV irradiation of living cells is widely used to identify protein–RNA interactions, and it is referred to as the crosslinking and immunoprecipitation (CLIP) method [35]. Owing to its accuracy in detecting genuine in vivo binding sites of an RBP [35], CLIP has become the gold standard to analyze protein–RNA interactions [15,32]. In the CLIP method, protein–RNA interactions are UV-crosslinked in living cells, followed by RIP (Figure 2). Following the lysis of the UV-crosslinked cells with appropriate lysis buffer, DNA in the cell lysates is eliminated with DNase, and RNA is fragmented with a low concentration of RNase. Then, the crosslinked-protein–RNA complexes are immunoprecipitated with a specific antibody against an RBP of interest and extensively washed on beads with high-salt buffer containing detergents to remove non-specifically remaining RNA and proteins. To further purify the target protein–RNA complexes, the immunoprecipitants are separated on SDS-polyacrylamide gel (SDS-PAGE) and transferred to a nitrocellulose membrane, which captures protein–RNA complexes but not free RNA. On the membrane, the protein–RNA complexes migrate above the molecular weight (MW) of the target RBP. The RNA retaining the short segments (≈50 nt) is isolated from the membrane and amplified by RT-PCR to generate a cDNA library. Although the multiple purification steps of a trace amount of immunoprecipitated RNA fragments substantially complicate the protocol and the required amount of input materials, these steps greatly enhance the signal-to-noise ratio of purified RNA fragments compared to the traditional RIP method.

The initial CLIP method employed cDNA cloning followed by Sanger sequencing to determine the identity of each isolated RNA [35]. Although hundreds of sequencings revealed some of the binding sites of Nova, the neuronal splicing factor in pre-mRNA, a large part of its binding sites remained unelucidated. The development of high-throughput sequencing technology has drastically improved this situation. High-throughput sequencing of the cDNA library generated by CLIP (CLIP-seq) produced robust transcriptome-wide maps of Nova–RNA interaction sites in the mouse brain [36]. The analysis demonstrated that the binding position of Nova in pre-mRNA is linked to the outcome of alternative splicing as well as that of alternative polyadenylation. The initial CLIP-seq mapped RBP-binding footprint regions at a resolution of 30–60 nucleotides, depending on the size of the fragmented RNA. Later, the identification of the crosslinked sites at single nucleotide resolution was achieved by several improvements, including the evaluation of crosslink-induced mutation sites (CIMS), which are nucleotide deletions or substitutions introduced at the protein–RNA crosslink sites by reverse transcriptase [37,38,39] and the detection of crosslink-induced truncation sites (CITS), which are generated by stopping the reverse transcription at crosslinked nucleotides [40].

Currently, the datasets of hundreds of CLIP-seq and its variations are available in public databases, including enhanced CLIPs (eCLIPs) of 150 RBPs in the Encyclopedia of DNA Elements (ENCODE) project [41]. These analyses have led to a deep insight into the relationship between the binding position of an RBP in pre-mRNA and the regulation of mRNA processing. The binding sites of RBPs are generally enriched around alternative splice sites rather than the constitutive splice sites. The serine/arginine-rich splicing factors (SRSFs) typically bind to exons to promote splicing, whereas the binding of heterogeneous ribonuclear proteins (hnRNPs) is mostly associated with exon skipping. Additionally, several RBPs display a position-specific regulation of alternative splicing, in which the proximal intronic binding of an RBP upstream and downstream of an exon suppresses and enhances exon inclusion, respectively. Similar position-specific binding of RBPs to pre-mRNA has also been observed in the regulation of alternative polyadenylation [5,42,43,44,45,46]. Furthermore, the integrated analysis of CLIP data and in vivo RNA structural data using PrismNet (Protein-RNA Interaction by Structure-informed Modeling using deep neural NETwork) revealed that local RNA structures dynamically regulate RBP-bindings in various cellular conditions [47].

The application of CLIP technology is not limited to the detection of RBP-binding sites in RNA. CLIP has been adopted to detect methyl adenosine 6 (m6A) modification sites [48,49] as well as secondary structure sites [50]. Furthermore, the combined analysis of CLIP and mass spectroscopy has been used to determine the precise regions of a protein that contact RNA [51,52,53].

### 2.3. Enhancement of the Sensitivity of the CLIP Methodology

Despite the advances in CLIP technology, the identification of protein–RNA interaction sites specifically assembled on transcribing RNAP II is still challenging, which is mainly because of the limited amount of nascent RNA associated with transcribing RNAP II machinery. The CLIP methodology requires a large amount of input materials, typically millions of cells, due to the loss of materials during multiple purification steps [32]. Infrared-CLIP (irCLIP) is an improved method that directly minimizes the loss of low-abundance materials [54]. Streamlining several steps in the CLIP protocol (Table 1), including the use of infrared-dye-conjugated RNA adapter for the visualization of the fragmented RNA, and the use of thermostable group II intron reverse transcriptase (TGIRT) for cDNA synthesis, has allowed the productive sequencing of cDNA libraries from 20,000 cells. eCLIP technique also involves the optimization of reaction conditions in RNA and cDNA handling (Table 1), which decreases the requisite PCR-amplification by ≈1000-fold in the cDNA library generation as compared with iCLIP [55], although most eCLIPs were performed using tens of millions of cells [41].

We recently reported the target RNA immunoprecipitation (tRIP) method [56], in which UV-crosslinked protein–RNA complexes are treated with the Terminator 5′-phosphate-dependent exonuclease (TEX) instead of purification with SDS-PAGE and membrane transfer in CLIP (Table 1). As TEX-dependent digestion of RNA is stopped at a protein-tethered nucleotide, the TEX treatment retains protein-crosslinked RNA but eliminates linkers and non-specific RNA. In addition, a cDNA library is generated by a single RNA purification step, similar to single-cell RNA-seq analysis [57]. These modifications simplify the experimental protocol and remarkably increase the sensitivity, while having the specificity comparable to that of CLIP. The tRIP further decreased the requisite PCR-amplification by ≈100-folds in the cDNA library generation, compared with the improved CLIP method, eCLIP, and identified protein–RNA interaction sites from thousands of cells.

The high sensitivity of tRIP-seq enabled the identification of protein–RNA interaction sites assembled on RNAP II by serial IP of RNAP II–RNA–protein complexes (Figure 3) [56]. Following UV-crosslinking, the RNAP II machinery, including associated RNA and proteins, is immunoprecipitated from cell lysates using an antibody specific to RNAP II. Next, the immunoprecipitants are treated with a low concentration of RNase to release protein–RNA complexes, which are further immunoprecipitated with an antibody against a protein of interest. Then, a tRIP library is generated, and high-throughput sequencing is performed to identify the protein binding sites specifically assembled on the RNAP II machinery. The application of the method to FUS, the RBP causally associated with amyotrophic lateral sclerosis (ALS) [58], showed that FUS binds upstream of alternative polyadenylation (APA) sites of the nascent RNA bound to RNAP II, to repress APA. Such migration of FUS was not observed in FUS-tRIP using whole cell lysates or immunoprecipitated spliceosome machinery. Thus, tRIP-seq characterizes the interactions of nascent RNA with RBPs specific to the RNAP II machinery. The results were obtained from meta-gene analysis, and the detected interaction sites were distributed sparsely at the individual gene level, suggesting that higher sensitivity is required for the precise analysis of co-transcriptional RNA processing.

Recently, a pulsed femtosecond UV to CLIP methodology (kinetic cross-linking and immunoprecipitation; KIN-CLIP) has been reported [59]. Compared with conventional UV irradiation, high-intensity UV irradiation markedly increases the crosslinking efficiency without altering the RNA–protein crosslinking patterns. Furthermore, the shortened irradiation time minimizes protein/RNA damage and enables the evaluation of binding and dissociation kinetics of protein–RNA interactions. The KIN-CLIP revealed that DAZL, the essential RBP for male and female gametogenesis, resides at individual binding sites for time periods of only a few seconds, whereas the binding sites remain DAZL-free for much longer periods. Utilizing the high sensitivity of KIN-CLIP is a promising way to identify dynamically altering protein–RNA interactions during transcription.

## 3. Other Strategies to Detect Protein–RNA Interactions in Living Cells

In addition to the CLIP technology, other strategies that do not rely on crosslinking have been explored for the detection of protein–RNA interactions in vivo.

### 3.1. The Utilization of RNA-Editing Mechanism

A novel technique known as TRIBE (targets of RNA-binding proteins identified by editing) utilizes an RNA-editing mechanism to identify protein–RNA interactions in a small number of specific cells [60]. This method expresses a fusion protein of an RBP and the catalytic domain of the RNA-editing enzyme, ADAR, in the cells of interest (Figure 4). ADAR deaminates adenosine to inosine, which is read by reverse transcriptase as guanosine [61]. As the RNA-binding domain of ADAR is replaced with the RBP of interest, target mRNAs of the RBP are marked with novel RNA editing events and identified by RNA-seq. TRIBE successfully identified the target mRNAs of an RBP from as few as 150 specific fly neurons. However, TRIBE-mediated editing is quite selective and probably gives rise to a high false-negative rate of identification of RBP-targeted RNAs [62]. The edited sites are strongly enriched in double-stranded regions of RNA, reflecting the preference of endogenous ADAR proteins [60,62]. Furthermore, these sites are diffusely distributed within hundreds of nucleotides apart from the RBP-binding sites detected by CLIP [60,62], suggesting the need for further improvements to adopt TRIBE, to specify protein–RNA interaction sites in RNA at high resolution.

Quite recently, a new method, STAMP (Surveying Targets by APOBEC Mediated Profiling), has been developed [63]. For the detection of protein–RNA interactions, STAMP uses the fusion protein of RBP and APOBEC1, which is a cytosine deaminase that edits RNA cytosine to uracil (Figure 4). When STAMP was performed with polyA+ mRNA-seq using a substantial number of cells (typically those from a 10 cm dish), the enrichment of edited sites was observed within the RBP-clustering regions detected in eCLIP. Furthermore, STAMP with single-cell RNA-sequencing (scRNA-seq) detected target mRNAs of an RBP at the single-cell level, showing extremely high sensitivity to STAMP. As both mRNA-seq and scRNA-seq limit the STAMP analysis to RBP-binding sites within exonic regions, the sites within intronic regions, which play a key role in the regulation of mRNA processing, are mostly missed in the current STAMP protocol. The adaptation of targeted sequencing of nascent RNA, such as native elongating transcript sequencing (NET-seq) [64,65], is an anticipated strategy to detect RBP-binding events specific to co-transcriptional RNA processing.

### 3.2. The Utilization of the Proximity Labeling System

Proximity labeling (PL) is an efficient tool for elucidating the protein–protein interactome within a specific subcellular fraction. The genetic fusion of the PL enzymes, such as APEX [66], BioID [67], or TurboID [68], to a protein of interest enables biotin labeling of neighboring proteins, which are pulled down with streptavidin beads and identified by mass spectrometry. The short half-lives of reactive species generated by the PL enzymes limit the labeling of the proteins to the near vicinity (typically 1–10 nm) in living cells.

In addition to the analysis of protein–protein networks, PL has been used to profile protein–RNA interactions. The combination of PL with crosslinking of protein–RNA interactions by UV (Proximity-CLIP) [69] or formaldehyde (APEX-RIP) [70] enabled the collection of RNAs bound to the biotin-labeled proteins. The identities of the isolated RNAs and proteins were determined using RNA-seq and mass spectrometry, respectively. Additionally, recent studies have developed the direct labeling of RNA by PL enzymes. In APEX-seq [71], the peroxidase enzyme APEX2 [72], which is fused with a protein of interest, directly biotinylates RNA in close proximity. In chromophore-assisted proximity labeling and sequencing (CAP-seq) [73], the light-activated singlet oxygen generator, miniSOG, is utilized for proximity oxidation of RNA, which can be captured by amine probes. These methods have successfully elucidated specifically enriched RNA molecules, especially in the membrane compartments, such as mitochondria, nucleus, cytoplasm, endoplasmic reticulum, and cell–cell junctions. In addition, Proximity-CLIP detected the RBP-protected footprints on mRNA, although the identities of the bound proteins were missed and presumed with the nucleotide motifs enriched in the footprinted regions [69]. Compared with CLIP-methodology, the simplicity of the protocol for collecting target RNA molecules offers a great advantage to PL-based methods. However, further improvements are warranted to identify protein–RNA interaction sites at high resolution, particularly in the non-membranous assembly, RNAP II.

## 4. Conclusions

Protein–RNA interactions are essential for a wide range of cellular processes, and various approaches have been developed to identify the RBP binding sites in RNA. Among these, CLIP remains a standard method owing to its accuracy in detecting the genuine interaction sites in vivo. However, CLIP requires a large amount of input materials, making it difficult to analyze the protein–RNA interactions specific to co-transcriptional RNA processing. Several improved and specialized versions, including irCLIP, eCLIP, tRIP, and KIN-CLIP, have greatly enhanced the detection sensitivity of CLIP. tRIP has enabled the identification of the protein–RNA interactions specifically assembled on transcribing RNAP II by meta-gene analysis, although the distribution of the detected interaction sites was sparse at the individual gene level. Recently, other strategies that utilize RNA editing or PL mechanisms have been developed for the detection of protein–RNA interactions. These methods have their distinct advantages and disadvantages compared with CLIP (Table 2). PL is advantageous in the isolation of proteins and RNA within a subcellular fraction. The combination of RNAP II-specific PL with the improved CLIP method or RNA editing strategy is anticipated to greatly enhance the detection efficiency of protein–RNA interactions assembled on RNAP II. Although further methodological improvements are warranted, the precise analysis of protein–RNA interactions during co-transcriptional RNA processing may be enabled in the near future.

## Figures and Tables

**Figure 1 ijms-22-05312-f001:**
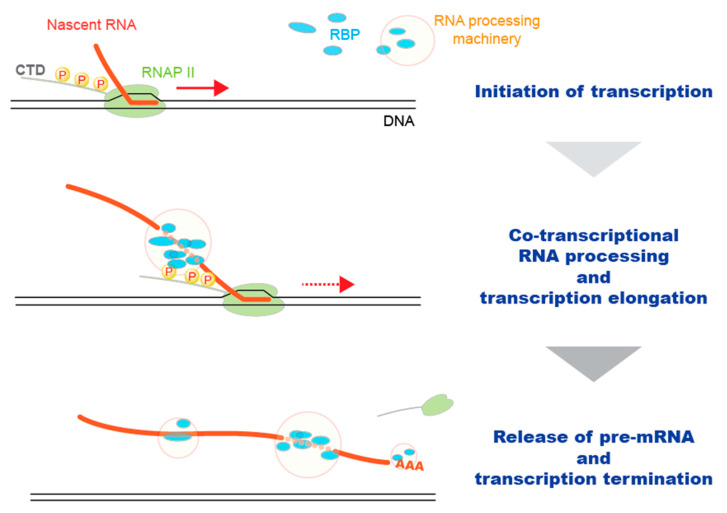
Transcription and co-transcriptional RNA processing. RNA processing machinery is recruited to the transcribing RNAP II machinery depending on the phosphorylation status of C-terminal domain (CTD). Then, protein–RNA interactions are formed in the recruited machinery to process nascent RNA co-transcriptionally. Along with transcription termination and polyadenylation, pre-mRNA is released from the RNAP II.

**Figure 2 ijms-22-05312-f002:**
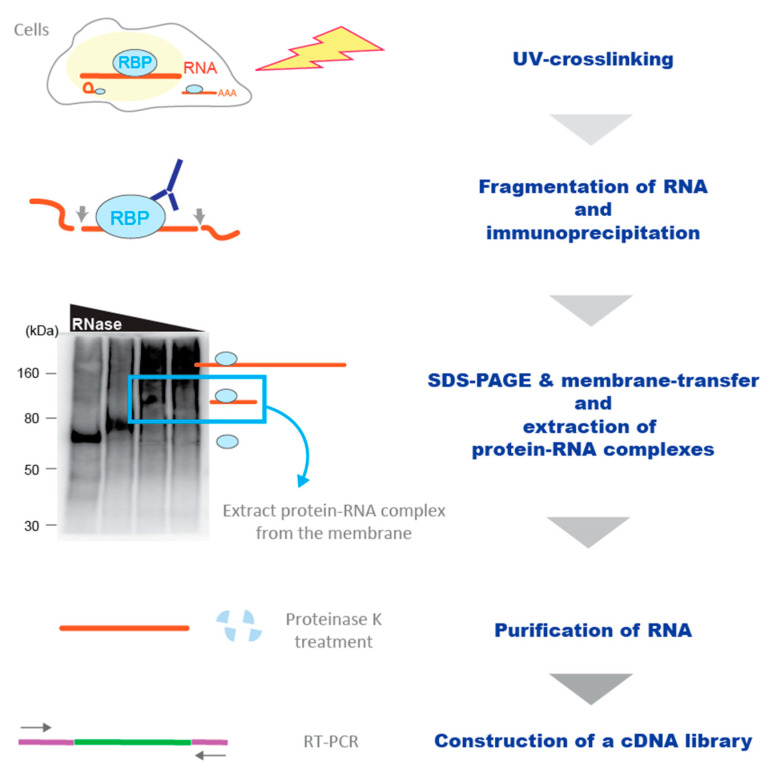
Schematic representation of the CLIP workflow. RNA-binding protein (RBP)–RNA interactions are UV-crosslinked in living cells. Following the lysis of cells, RNA is partially digested with RNase, and the crosslinked complex is immunoprecipitated with an antibody against an RBP of interest. After stringent washes, the immunoprecipitants are separated on SDS-PAGE and transferred to nitrocellulose membrane, on which the RBP complexed with RNA migrate above the MW of the target RBP. Then, the complexes retaining the short segments (≈50 nt) are isolated from the membrane. Following the proteinase K-treatment and RNA purification, a cDNA library is constructed with RT-PCR.

**Figure 3 ijms-22-05312-f003:**
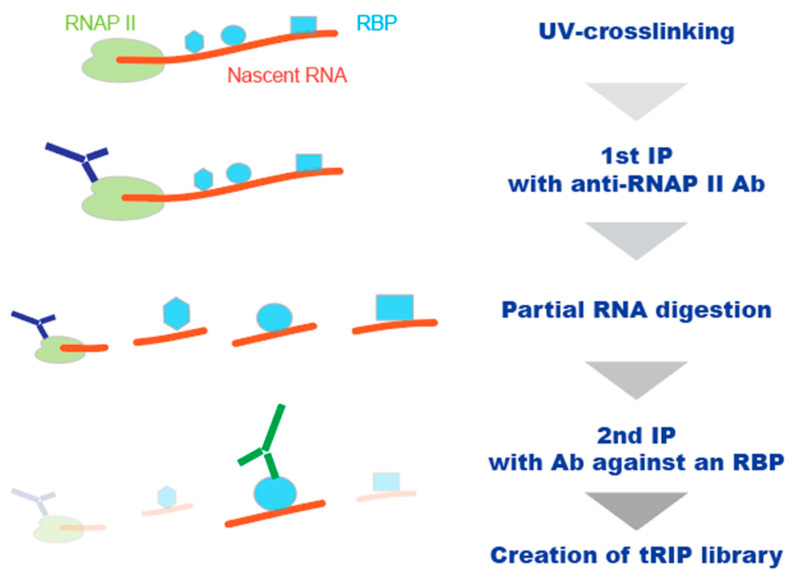
Schematic representation of the tRIP analysis of protein–RNA interactions specifically assembled on RNAP II. Following UV-crosslinking, protein–RNA complexes were immunoprecipitated with an anti-RNAP II antibody (1st IP). RNA of the isolated RNAP II–RNA–RBP complexes was partially digested with RNase, and the released RBP–RNA complexes were further immunoprecipitated with antibody against an RBP of interests (2nd IP). Then, the immunoprecipitated RBP–RNA complexes are subjected to tRIP-seq analysis.

**Figure 4 ijms-22-05312-f004:**
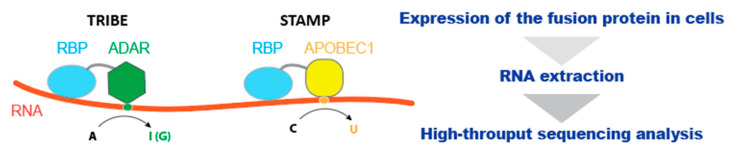
Schematic representation of TRIBE (targets of RNA-binding proteins identified by editing) and STAMP (surveying targets by APOBEC mediated profiling) strategies. A fusion protein of an RBP and an RNA editing enzyme edits the nucleotides around RBP-binding sites. In TRIBE, an RBP is fused to the catalytic domain of ADAR that catalyzes an adenosine-to-inosine conversion. In STAMP, it is fused to APOBEC1 that catalyzes a cytosine-to-uracil conversion. The edited sites can be detected using high-throughput RNA sequencing analysis.

**Table 1 ijms-22-05312-t001:** Comparison of irCLIP, eCLIP, and tRIP protocols with CLIP protocol.

	CLIP	irCLIP	eCLIP	tRIP
UV crosslinking	√	√	√	√
Cell lysis and IP	√	√	√	√
RNase treatment	Total cell lysates	on beads	Total cell lysates	on beads
Dephosphorylation of RNA	√	√	√	–
5′ end labeling	√	Replaced with infrared dye-labeling of 3′ linker	√	–
3′ linker ligation	√	√	√	√
SDS-PAGE	√	√	√	Replaced with deadenylase- and TEX-treatments
Transfer to membranes	√	√	√	
Cut membrane	√	√	√	
Proteinase K treatment	√	√	√	√
Purification of RNA	Phenol/chloroform	Phenol/chloroform	Phenol/chloroform	Column purification
	ETOH precipitation	ETOH precipitation	Column purification	–
Reverse transcription	√	√	√	√
Purification of cDNA/RNA	–	Pull-down with streptavidin beads	Silane-beads purification	–
Modification of 5′ end	5′ linker ligation	Circularization of cDNA	5′ linker ligation	PolyA tailing
Purification of cDNA/RNA	Phenol/chloroform ETOH precipitation	Silane-beads purification	Silane-beads purification	–
PCR amplification	√	√	√	√
Hands-on time	4 days	3 days	4 days	2 days

**Table 2 ijms-22-05312-t002:** The advantages and disadvantages of the methods to investigate protein–RNA interactions.

Method	Advantages	Disadvantages	Analysis of the Interactions Specific to the RNAP II Machinery
**RIP**	Well-studied Easy-to-use	Dependent on antibody specificity Low signal-to-noise ratio Requires substantial amounts of input materials Does not determine the exact location of RBP-binding sites	Difficult to isolate RNAP II fraction keeping physiological protein–RNA interactions
**CLIP** irCLIP [54] eCLIP [55] tRIP [56]	Identification of direct protein–RNA interaction sites at single nucleotide resolution High signal-to-noise ratio	Dependent on antibody specificity Low crosslinking efficiency Requires substantial amounts of input materials Complicated procedures	tRIP succeeded in the RNAP II-specific detection Requires further enhancement of detection sensitivity for the precise analysis
KIN-CLIP [59]	High crosslinking efficiency	Requires dedicated devices	Not examined. Requires optimization for less input materials
**RNA editing** TRIBE [60] STAMP [63]	No need to purify protein–RNA complexes No dependence on crosslinking High detection sensitivity (from single-cell level)	Requires artificial expression of an RBP fused with an RNA-editing enzyme The distribution of detected sites is biased, reflecting the preference of the fused RNA-editing enzyme	Not examined Requires isolation or labeling of RNA specific to the RNAP II machinery
**Proximity labeling** Proximity-CLIP [69] APEX-RIP [70] CAP-seq [73]	No dependence on crosslinking Efficient isolation of RNA in a specific subcellular fraction	Requires artificial expression of a PL enzyme specific to the fraction of interest Does not identify specific binding sites of an RBP of interest	Not examined Requires development of the labeling strategy specific to the RNAP II machinery

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
