# Peer review of "Rapidly Growing Protein-Centric Technologies to Extensively Identify Protein–RNA Interactions: Application to the Analysis of Co-Transcriptional RNA Processing"

_ijms, 2021, doi:10.3390/ijms22105312_

Round 1

Reviewer 1 Report

RNA-protein interaction is crucial for most biological processes in all organisms. The authors summarized and compared several methods that were used to identify the protein-RNA interactions.

Comments:

  1. The authors should provide a detailed future perspective on this topic and how it can be implemented to be taken forward these method developments.
  2. Many abbreviations in the manuscript affect readers' understanding the manuscript. The authors should provide their full names and briefly define them.
  3. The authors should cite these articles and describe the differences from their manuscript. “Hafner M, Katsantoni M, Köster T, Marks J, Mukherjee J, Staiger D, Ule J, Zavolan M. CLIP and complementary methods. Nature Reviews Methods Primers. 2021 Mar 4;1(1):1-23.” “Popova VV, Kurshakova MM, Kopytova DV. [Methods to study the RNA-protein interactions]. Mol Biol (Mosk). 2015 May-Jun;49(3):472-81. Russian. doi: 10.7868/S0026898415020111. PMID: 26107901.”
  4. The authors should mention the new method. For example, Sun, Lei, Kui Xu, Wenze Huang, Yucheng T. Yang, Pan Li, Lei Tang, Tuanlin Xiong, and Qiangfeng Cliff Zhang. "Predicting dynamic cellular protein–RNA interactions by deep learning using in vivo RNA structures." Cell research (2021): 1-22.
  5. Line 52, remove duplicate “and hundreds”.
  6. Line 110-121 should be merged into section 2.2.

Author Response

We greatly appreciate the productive and encouraging comments by the reviewer, which have helped to improve our manuscripts.  The revisions are highlighted in the manuscript.

Point 1: The authors should provide a detailed future perspective on this topic and how it can be implemented to be taken forward these method developments.

 Response 1: Thank you for the insightful comment. We revised the conclusion section and added a new table (Table 2) showing the advantages and disadvantages of the presented methods. In the table, we also speculated current problems in the application of these methods to the analysis of protein-RNA interactions specific to the RNAP II machinery.

Point 2: Many abbreviations in the manuscript affect readers' understanding the manuscript. The authors should provide their full names and briefly define them.

Response 2: We apologize for the complicated usage of abbreviations. We confirmed that each abbreviation is spelled out the first time it appears in the manuscript. In addition, we have added a list of abbreviations to increase readability (Supplementary Table 1).

Point 3: The authors should cite these articles and describe the differences from their manuscript. “Hafner M, Katsantoni M, Köster T, Marks J, Mukherjee J, Staiger D, Ule J, Zavolan M. CLIP and complementary methods. Nature Reviews Methods Primers. 2021 Mar 4;1(1):1-23.” “Popova VV, Kurshakova MM, Kopytova DV. [Methods to study the RNA-protein interactions]. Mol Biol (Mosk). 2015 May-Jun;49(3):472-81. Russian. doi: 10.7868/S0026898415020111. PMID: 26107901.”

Response 3: Thank you for the suggestions. We have referred to the indicated articles and stated the focus of our manuscript in the introduction section, to show the differences from these articles. We have also revised the title to indicate the focus more clearly.

Point 4: The authors should mention the new method. For example, Sun, Lei, Kui Xu, Wenze Huang, Yucheng T. Yang, Pan Li, Lei Tang, Tuanlin Xiong, and Qiangfeng Cliff Zhang. "Predicting dynamic cellular protein–RNA interactions by deep learning using in vivo RNA structures." Cell research (2021): 1-22.

Response 4: According to the reviewer’s suggestion, we have referred to the indicated article in section 2.2.

Point 5: Line 52, remove duplicate “and hundreds”.

Response 5: We apologize for the mistake. We corrected it.

Point 6: Line 110-121 should be merged into section 2.2.

Response 6: According to the reviewer’s comment, we edited lines 110-121 and section 2.2.

Reviewer 2 Report

Masuda et al. present a short review on the method to detect RNA-protein interactions in cells.

In general the review is well-organised and informative. I would however say that the title is misleading. The title promises a review of technologies used to identify RNA-protein interactions. However, the only technologies which are mentioned are cellular based CLIP or RIP. A very important part of the methodological landscape, which relies on selected RNA-centered pull down methods in cell or cell extract, is omitted. Additionally, a description of methodological papers focused on identifying known and new proteins binding to RNA - such as RBPmap, RNA interactome capture etc. is not presented in the review. I see two solutions: 1) to change the title and make is much more focused, 2) to expand the review and include these other methods.

Another concept lacking depth from the review is critical analysis. All presented methods have their advantages and disadvantages, which should be better presented or summarised in a table. For example, generating fusion proteins with ADAR or APOBEC1 can influence specificity, protein partners and function of fused RBPs.

Finally, I have spotted one mistake:

lane 52 - 'and hundreds' should be deleted

Author Response

We profoundly thank this reviewer for precise understanding of our study and giving us comments that improved our manuscript. The revisions are highlighted in the manuscript.

Point 1: The title promises a review of technologies used to identify RNA-protein interactions. However, the only technologies which are mentioned are cellular based CLIP or RIP. A very important part of the methodological landscape, which relies on selected RNA-centered pull down methods in cell or cell extract, is omitted. Additionally, a description of methodological papers focused on identifying known and new proteins binding to RNA - such as RBPmap, RNA interactome capture etc. is not presented in the review. I see two solutions: 1) to change the title and make is much more focused, 2) to expand the review and include these other methods.

Response 1: As the reviewer sharply pointed out, the current manuscript mainly describes the protein-centric methods to identify endogenous protein-RNA interactions. According to the reviewer’s suggestion, we have revised the title.

Point 2: Another concept lacking depth from the review is critical analysis. All presented methods have their advantages and disadvantages, which should be better presented or summarised in a table.

Response 2: Thank you for the insightful comment. We have added a new table showing the advantages and disadvantages of the presented methods (Table 2).

Point 3: lane 52 - 'and hundreds' should be deleted.

Response 3: We apologize for the mistake. We corrected it.

Round 2

Reviewer 1 Report

no new comments

Reviewer 2 Report

Authors addressed my concerns about the title and the critical analysis of presented methods. The manuscript is ready for publication.